# Preservation of Smooth Muscle Cell Integrity and Function: A Target for Limiting Abdominal Aortic Aneurysm Expansion?

**DOI:** 10.3390/cells11061043

**Published:** 2022-03-19

**Authors:** Emily R. Clark, Rebecca J. Helliwell, Marc A. Bailey, Karen E. Hemmings, Katherine I. Bridge, Kathryn J. Griffin, D. Julian A. Scott, Louise M. Jennings, Kirsten Riches-Suman, Karen E. Porter

**Affiliations:** 1Institute of Medical and Biological Engineering (iMBE), School of Mechanical Engineering, University of Leeds, Leeds LS2 9JT, UK; emilyclark.erc@gmail.com (E.R.C.); l.m.jennings@leeds.ac.uk (L.M.J.); 2Multidisciplinary Cardiovascular Research Centre (MCRC), University of Leeds, Leeds LS2 9JT, UK; m.a.bailey@leeds.ac.uk (M.A.B.); k.e.hemmings@leeds.ac.uk (K.E.H.); k.i.bridge@leeds.ac.uk (K.I.B.); k.j.griffin@leeds.ac.uk (K.J.G.); d.j.a.scott@leeds.ac.uk (D.J.A.S.); 3Leeds Institute of Cardiovascular and Metabolic Medicine (LICAMM), University of Leeds, Leeds LS2 9JT, UK; um13rjh@leeds.ac.uk (R.J.H.); k.riches@bradford.ac.uk (K.R.-S.); 4Leeds Vascular Institute, Leeds General Infirmary, Leeds LS1 3EX, UK; 5Department of Histopathology, St James University Hospital, Leeds LS9 7TF, UK; 6Cardiovascular Research Group, Faculty of Life Sciences, University of Bradford, Bradford BD7 1DP, UK

**Keywords:** abdominal aortic aneurysm, bioreactor, tissue strength, smooth muscle cell phenotype, proliferation, senescence, MMP-2, bystander effect

## Abstract

(1) Abdominal aortic aneurysm (AAA) is a silent, progressive disease with significant mortality from rupture. Whilst screening programmes are now able to detect this pathology early in its development, no therapeutic intervention has yet been identified to halt or retard aortic expansion. The inability to obtain aortic tissue from humans at early stages has created a necessity for laboratory models, yet it is essential to create a timeline of events from EARLY to END stage AAA progression. (2) We used a previously validated ex vivo porcine bioreactor model pre-treated with protease enzyme to create “aneurysm” tissue. Mechanical properties, histological changes in the intact vessel wall, and phenotype/function of vascular smooth muscle cells (SMC) cultured from the same vessels were investigated. (3) The principal finding was significant hyperproliferation of SMC from EARLY stage vessels, but without obvious histological or SMC aberrancies. END stage tissue exhibited histological loss of α-smooth muscle actin and elastin; mechanical impairment; and, in SMC, multiple indications of senescence. (4) Aortic SMC may offer a therapeutic target for intervention, although detailed studies incorporating intervening time points between EARLY and END stage are required. Such investigations may reveal mechanisms of SMC dysfunction in AAA development and hence a therapeutic window during which SMC differentiation could be preserved or reinstated.

## 1. Introduction

Abdominal aortic aneurysm (AAA) is an asymptomatic disease characterised by progressive dilatation of the abdominal aorta that, without intervention, culminates in rupture that carries a mortality of 65–80% [1]. Presenting typically in males over the age of 65 years [2], AAA prevalence in the UK is reportedly 1.3%, yet no pharmacological or other therapeutic agents are available to treat or halt progression [3,4]. With routine surveillance commencing after detection, it is generally accepted that once an AAA exceeds 5.5 cm in diameter, risk of rupture outweighs the risk associated with reparative surgery, the only treatment option [5]. Despite decades of research, AAA diameter is still the only approved metric for predicting rupture [6].

AAA pathophysiology carries several hallmark features: inflammation [7,8]; loss of extracellular matrix (ECM) associated with aberrant matrix metalloproteinase activity (MMPs) [9,10,11]; and, crucially, dysfunction and death of the principal cell type of the aortic wall—vascular smooth muscle (SMC) [12,13]. Over time, these processes lead to gradual weakening of the aortic wall and resultant devastating loss of its biomechanical properties [14,15,16]. It is therefore essential to construct a temporal view of events in order to seek targets for intervention at an earlier stage of disease when therapeutic agents might improve AAA outcomes.

SMC are inherently plastic and not terminally differentiated: they reversibly switch from a differentiated, contractile phenotype to a de-differentiated, secretory phenotype according to external mechanical and biochemical cues [17,18]. SMC de-differentiation is characterised by downregulation of SMC differentiation markers such as alpha-smooth muscle actin (α-SMA) and smooth muscle myosin heavy chain (SM-MHC) [19]. Such plasticity aids in SMC-mediated vascular repair following injury [20]. In AAA, a unique SMC phenotype has been characterised by our group and others [21,22,23]. Pathway analysis of the meta-analysis of AAA genome-wide association studies identified “abnormal vascular smooth muscle cell physiology” amongst the gene enrichment set from the top 10 validated loci [24]. AAA-SMC can degrade insoluble elastin, have increased levels of senescence, possess rhomboid morphology, and exhibit altered proliferative capacity. SMC are also responsible for ECM synthesis, linking AAA-SMC with the characteristic altered ECM in AAA [21,23,25].

A major obstacle to AAA research in humans is an inability to obtain human tissue in the early stages of the disease which might provide clues as to the mechanisms that provoke the progressive destruction of aortic integrity. Animal models have therefore formed the basis of in vivo studies to explore the aetiology of early AAA development. Small mammal models such as those using genetic deletion of ApoE are widely employed [26,27]; in other studies, application of elastase to the abdominal aorta to induce AAA-like dilation [28,29] has been documented. This latter approach has been translated to large animals whose vascular physiology is more compatible with humans [30,31,32]. In our own laboratories, we developed and validated a novel ex vivo model of porcine arteries pre-treated with protease enzymes and subsequently cultured under flow conditions with retained viability in a bioreactor for 12 days [23]. Vessels were retrieved and examined histologically, and cultures of SMC were also established, with a key outcome that unambiguous vessel wall disruption (identified histologically) and SMC aberrancies compatible with human end stage disease were present [23]. In a second study, we therefore introduced a shorter (3 day) time interval for bioreactor culture in search of a potential time point at which critical cellular changes are less advanced or undetectable [33].

The aims of the current study were twofold. Firstly, to study biomechanical properties of 12 day porcine vessels (END stage). Secondly, to perform pertinent assays of function and phenotype in parallel, on SMC derived from 3 days (EARLY-SMC) and 12 days (END-SMC) to investigate whether there were any key differences that could be further explored as possible targets.

## 2. Materials and Methods

### 2.1. Bioreactor Model

Female 65 kg pigs were sedated with Stresnil (Elanco Animal Health, Hampshire, UK), anaesthetised with Hypnovel (Hoffman La Roche, Basel, Switzerland), and terminated via Pentoject (Animalcare, Yorkshire, UK) injection under Schedule 1 of the Animals (Scientific Procedures) Act 1986. All procedures complied fully with UK Home Office regulations. Porcine hair was removed, and the skin was treated with chlorhexidine for sterility and the left and right carotid arteries were dissected and removed under aseptic conditions in a dedicated large-animal operating theatre.

The configuration of the bioreactor and vessel preparation was as described previously [23] (Figure 1a,b). Briefly, collagenase (Worthington Biochemical, Lakewood, NJ, USA) and elastase (MP Biomedicals, Strasbourg, France) (3 mg/mL and 1.5 mg/mL final concentrations, respectively) were incorporated into an agar gel (0.5% final *w*/*v* in HBSS) which was applied peri-adventitially to aseptically harvested porcine carotid arteries (CCE). SMC explant culture and tissue rings fixed in 10% formalin were taken from freshly isolated tissue (FRESH). The contralateral carotid artery was treated with the agar gel containing no enzymes and thus served as a matched vehicle control (VEH). Vessels were incubated for 3 h prior to removal of the treatment with a media wash and installation in the bioreactor. The bioreactor was perfused with medium containing 30% foetal bovine serum at ≈120 mL/min^−1^ in 5% CO_2_ *v*/*v* O_2_ at 37 °C. Culture for 3 days represented early stage disease (EARLY), whereas culture for 12 days represented end-stage disease (END). Following this, the vessel was removed from the bioreactor and tissue samples were taken for SMC explant culture and either immunohistochemistry or uniaxial tensile testing. For both 3 and 12 day bioreactor experiments, there were three biological replicates where vessels were isolated from three separate animals.

### 2.2. Uniaxial Tensile Testing

For each porcine experiment, FRESH tissue was set up for mechanical testing at the time of harvesting. Twelve day cultured arteries (VEH and CCE) were recovered from the bioreactor and immediately set up for testing; each experiment thereby generated three sets of data. Central tissue strips (3 mm wide by 6 mm long) were dissected from each porcine artery and three strips were cut from each orientation: longitudinal and circumferential (Figure 1c). The mean thickness of each tissue strip was calculated by measuring the tissue thickness at three places along the length of the strip using a digital micrometre. Soft tissue clamps were custom made with fine grade sandpaper to grip with minimal slippage. The tissue was mounted and tested to failure in an Instron^®^ 3365 Machine equipped with a 50 N load cell (Instron, High Wycombe, UK) at a constant displacement rate of 10 mm/min. Physiological parameters were maintained by performing all mechanical testing experiments in an Instron BioPuls bath filled with 1X PBS at 37 °C (Figure 1d). No preload or preconditioning protocols were used to prevent masking of inherent tissue properties by initiating the data collection at an arbitrary value rather than the true relaxed state. Load and displacement were recorded at 0.1 s intervals and exported to Bluehill software (Instron, UK, version 2).

Load (F) and displacement (𝓁) data were converted to engineering stress (σ) and engineering strain (ε) according to Equations (1) and (2):
σ = F/(t w)(1)
ε = ∆ 𝓁/𝓁_0_(2)
where t is the mean thickness of the unloaded test strip and w is the tissue strip width. The elastin phase stiffness, collagen phase stiffness, ultimate tensile strength, and transition strain were calculated using a custom program written in MatLab (MathWorks, Natick, MA, USA).

### 2.3. Histology

Following 3 or 12 days in the bioreactor, short (<5 mm) rings of artery were removed, fixed in formalin, and embedded in paraffin. Sections (5 µm) were stained with mouse monoclonal anti α-SMA clone 1A4 (1:400; Sigma-Aldrich, Dorset, UK) and Millers elastin as previously described [34]. Images were captured using a Zeiss AxioVision Imaging System (AxioCam HRc camera on an AxioImager Z.1 microscope, Cambridge Scientific Products, Watertown, MA, USA). Sections were taken from three animals at both EARLY and END time-points.

### 2.4. SMC Isolation and Culture

Porcine SMC were cultured either directly after tissue harvesting (FRESH) or following bioreactor treatment (VEH or CCE). SMC were explanted as we previously described [23]. Cells were isolated from three animals at both EARLY and END time-points.

Human SMC (hSMC) were explanted using the same method, from saphenous vein fragments obtained from patients undergoing coronary artery bypass grafting at Leeds General Infirmary, UK. The study had local ethical committee permission (CA01/040). Each donor provided informed, written patient consent, and the study conformed to the principles outlined in the Declaration of Helsinki. hSMC from four different patients were used in this study.

Both SMC and hSMC were maintained in full growth medium (FGM: Dulbecco’s modified Eagle’s medium (DMEM) supplemented with 10% foetal bovine serum, 1% L-glutamine, and 1% penicillin/streptomycin/fungizone) at 37 °C in 5% CO_2_ in air. Once confluent, cells were serially passaged using 0.25% Trypsin/EDTA. All experiments were conducted on SMC between passages 3–5.

### 2.5. Morphometric Analyses

SMC explanted from all vessels were seeded (2 × 10^5^ cells per 75 cm^2^ flask) and maintained in FGM for 96 h. Ten random fields of view were imaged from SMC from each condition (FRESH, VEH, and CCE from both EARLY and END vessels) at 100× magnification using brightfield microscopy. The circularity of 50 individual cells per experimental group were measured using ImageJ software (version 1.52j). Circularity is a dimensionless geometric descriptor where 1.0 denotes a perfect circle and values approaching zero indicate an increasingly elongated polygon.

### 2.6. Proliferation

SMC proliferation was determined as we described previously [23]. SMC were seeded (1 × 10^4^ cells per well in a 24-well plate) and maintained in FGM for 24 h to adhere. Serum-free medium was used to quiesce cells in G0 prior to conducting “day 0” counts, after which SMC were transferred back to FGM as a maximal proliferation stimulus. Quadruplicate cell counts (technical replicates) using Trypan blue and a haemocytometer were made on days 0, 2, and 4 with a media change on day 2. FRESH, VEH, and CCE-SMC were cultured in parallel from cells isolated from three animals at both EARLY and END time-points.

To assess the mitogenic potential of conditioned medium (CM) from end-stage SMC, VEH and CCE-SMC were seeded (7 × 10^4^ cells per well in 6-well plates) in FGM to establish for 24 h, then serum-depleted for 72 h. Subsequently, CM was collected, centrifuged to remove cell debris, snap-frozen in liquid nitrogen, and stored at −80 °C until required. CM was supplemented with 1% FCS before use. hSMC from 4 different patients were seeded (3 × 10^3^ per well in 96-well plates) in FGM and incubated overnight to establish before quiescing for 72 h in serum free medium (DMEM supplemented with 1% L-glutamine and 1% penicillin/streptomycin/fungizone). On days 0 and 2, hSMC were treated with 100 µL CM derived from either END VEH or END CCE-SMC. Images were captured daily between days 0–4 using an Incucyte ZOOM (Sartorius, Goettingen, Germany). Confluence (as a measure of proliferation) was calculated using in-built image analysis thresholding software (Incucyte ZOOM 2016A) and expressed as percentage change from day 0.

### 2.7. Senescence-Associated β-Galactosidase Assay

SMC senescence was determined using a commercial assay of β-galactosidase (Cell Signalling Technology, Danvers, MA, USA) as described previously [23]. Briefly, cells were seeded (7.5 × 10^4^ cells per well in a 6-well plate) and cultured for 48 h in FGM. The assay was performed according to manufacturer’s instructions and the presence of senescence-associated β-galactosidase at pH6 resulted in a blue precipitate that was detectable histochemically. Ten brightfield microscopic images (40× mag.) were captured from each well, and senescence score was calculated and normalised to FRESH. Senescence was measured in cells isolated from three animals at both EARLY and END time-points.

### 2.8. Gelatin Zymography

SMC were seeded (2 × 10^5^ cells per 25 cm^2^ flask) in FGM, cultured for 24 h to be established, and then quiesced in serum-free medium for 72 h. The cells were transferred to medium containing 0.4% FBS and either PBS control (0.01% *v*/*v*), TPA (100 nM), or PDGF-BB with IL-1α (10 ng/mL each; inflammatory stimulus) for a further 48 h. CM was collected from the cells, centrifuged to remove cell debris, snap-frozen in liquid nitrogen, and stored at −80 °C until required. Gelatin zymography was performed as described previously with a final gelatin concentration of 1 mg/mL [35]. CM from HT1080 sarcoma cells was used as a positive control for MMP-2 and MMP-9. A reference sample was included on each zymogram to enable comparison between gels. MMP secretion was measured in cells isolated from three animals at both EARLY and END time-points.

### 2.9. Statistical Analyses

All assays were conducted as technical triplicates on SMC from each animal in passage-matched experiments. Throughout the manuscript, *n* refers to the number of biological replicates (the number of animals from which SMC were isolated). All data were expressed as mean ± SEM. Non-parametric data (biomechanics, cell morphology) were analysed using a Kruskal–Wallis test with Dunn’s multiple comparison and a Mann–Whitney test with a Benjamini–Hochberg correction. Parametric data (cell proliferation, senescence, and MMP secretion) were analysed with repeated measures two-way ANOVA with Sidak’s multiple comparison post hoc test or paired *t*-test as appropriate. Differences between treatment groups (FRESH, VEH, CCE) and model stage (EARLY, END) were analysed using GraphPad Prism version 7 software. A *p*-value of <0.05 was considered statistically significant.

## 3. Results

### 3.1. Biomechanics of END Stage CCE Tissue

We previously demonstrated that END CCE-SMC are aberrant in phenotype and function [23,33]. The aim of the present experiment was to determine the possible impact of these defects on tissue integrity. Tissue strips in both the circumferential and longitudinal orientation were uniaxially tested at a constant displacement rate of 10 mm/min and the data were converted to engineering stress and strain. Elastin region stiffness, collagen region stiffness, transition strain, and ultimate tensile strength were calculated using a custom MatLab program.

For circumferential orientation, VEH tissue tended to decrease in stiffness compared to FRESH in both the elastin (0.16 ± 0.08 vs. 0.22 ± 0.01 MPa) and collagen regions (0.89 ± 0.32 vs. 1.46 ± 0.14 MPa) (Figure 1e,f). CCE tissue showed a further significant decrease in stiffness compared to FRESH in both the elastin (0.04 ± 0.02 MPa, *p* < 0.05) and collagen (0.24 ± 0.09 MPa, *p* < 0.05) regions. The difference in transition strain between FRESH and VEH tissue was not statistically significant (0.96 ± 0.07 vs. 0.78 ± 0.06 MPa).

There was a 40% decrease in transition strain in CCE compared to FRESH tissue (0.55 ± 0.10 MPa; Figure 1g). Although the ultimate tensile strength of VEH tissue was lower than FRESH, this was not significant (0.88 ± 0.21 vs. 1.44 ± 0.11 MPa, respectively). Furthermore, the ultimate tensile strength of CCE tissue was significantly lower than FRESH (0.30 ± 0.19 MPa, *p* < 0.05; Figure 1h). Ultimate tensile strength in FRESH tissue was fivefold greater than in CCE tissue.

In the longitudinal orientation, VEH tissue was less stiff compared to FRESH in both the elastin region (0.03 ± 0.01 vs. 0.16 ± 0.01 MPa) and the collagen region, although this was not statistically significant (0.21 ± 0.04 vs. 0.94 ± 0.25 MPa; Figure 1i,j). CCE tissue showed a further significant decrease in stiffness compared to FRESH in the elastin region (0.009 ± 0.002 MPa, *p* < 0.01) and collagen region (0.04 ± 0.01 MPa, *p* < 0.05; Figure 1i,j). There was no difference in transition strain between FRESH, VEH, or CCE tissue (Figure 1k). VEH tissue tended to be weaker than FRESH (0.45 ± 0.09 vs. 1.68 ± 0.24 MPa, respectively). Furthermore, CCE tissue tended to be weaker than VEH, and was significantly less than FRESH (0.06 ± 0.02 MPa, *p* < 0.01; Figure 1l).

Together, these data demonstrated that culturing any vessel for a period of 12 days in a bioreactor led to a reduction in both stiffness and strength (particularly in the longitudinal orientation) and that this was further exacerbated in the END CCE model.

### 3.2. Tissue and Cell Structure from EARLY and END Models

Histological analysis of EARLY tissues revealed that VEH and FRESH tissues were indistinguishable, each exhibiting distinct arterial layers with abundant α-SMA-positive cells and elastin in the media. The internal elastic lamina remained intact next to the lumen, and the adventitia was elastin-rich. Whilst EARLY CCE retained abundant α-SMA-positive cells, there was a generalised loss of elastin observed throughout the tissue (Figure 2a). In END tissues, VEH and FRESH again appeared indistinguishable; however, END CCE tissue presented with a generalised loss of elastin throughout the arterial layers (Figure 2a, closed arrows), and α-SMA staining was irregular (Figure 2a, open arrow).

We previously demonstrated that END CCE-SMC had significantly larger spread cell areas [23], displaying perinuclear granulation and a disorganised F-actin cytoskeleton, whilst EARLY CCE-SMC were indistinguishable from FRESH or VEH controls [33]. Increased spread cell area can indicate hypertrophy (where the cell maintains a similar shape but simply becomes larger) or a transition from a spindle-to-rhomboid morphology which is more indicative of phenotypic change [20]. Our current data confirmed that there was no change in EARLY CCE-SMC morphology compared to FRESH and VEH, both of which maintained a characteristic SMC spindle shape (FRESH 0.21 ± 0.01, VEH 0.22 ± 0.01, CCE 0.24 ± 0.01). However, END CCE-SMC were notably more rhomboid in nature than their FRESH or VEH counterparts (FRESH 0.21 ± 0.03, VEH 0.20 ± 0.003, CCE 0.52 ± 0.05; Figure 2b,c).

### 3.3. SMC Proliferation and Senescence

EARLY CCE-SMC had a significantly increased proliferation compared to either FRESH- or VEH-treated cells (11.2-fold increase compared to 9.3-fold in VEH; Figure 3a). In contrast, END CCE-SMC had a significantly impaired proliferation (7.4-fold increase compared with 12.2-fold in VEH; Figure 3b). We previously demonstrated that apoptosis in END CCE-SMC and VEH-SMC was comparable [33]; hence, we explored whether the perceived reduction in proliferation of CCE-SMC was due to the cells becoming senescent. We observed no change in the proportion of senescent cells in either FRESH, VEH, or CCE-SMC in the EARLY model. In contrast, culture of the VEH vessel in the bioreactor for 12 days led to a significant increase in SMC senescence (2.2-fold vs. FRESH SMC, *p* < 0.01) which was further increased in END CCE-SMC (3.3-fold vs. VEH SMC, *p* < 0.01; Figure 3c,d).

### 3.4. Matrix Remodeling

We previously demonstrated that END CCE-SMC secreted reduced levels of MMP-2 both under basal conditions and in response to phorbol ester (TPA) [23]. AAA is a hyper-inflammatory condition in which platelet-derived growth factor (PDGF) and interleukin-1 (IL-1) are both known to be increased [36,37].

Thus, we proceeded to investigate whether these factors could influence MMP-2 secretion in both EARLY and END stage SMC. In EARLY SMC, a combination of PDGF/IL-1 (each 10 ng/mL) induced MMP-2 secretion to a similar level as TPA (Figure 4a,b). However, cells from one animal consistently secreted less MMP-2 than the other two animals. As was the case previously [23], the END model CCE-SMC secreted significantly less MMP-2 than VEH-SMC. PDGF/IL-1 increased MMP-2 secretion in FRESH-SMC to a level comparable with TPA, although this induction was not observed in either VEH-SMC or CCE-SMC (Figure 4c,d).

### 3.5. Secretion of Paracrine Factors That Propagate SMC Dysfunction

In addition to metalloproteinase enzymes, it is likely that aneurysmal SMC secrete paracrine factors that can impact neighbouring, non-diseased SMC: a so-called “bystander” effect [38]. Naïve hSMC were treated with CM collected from VEH-SMC and CCE-SMC. In the EARLY model, CM from CCE-SMC induced cellular proliferation by 2.7-fold, not significantly different than CM from VEH-SMC (2.4-fold). In contrast, CM from CCE-SMC from the END model induced a threefold increase in proliferation in naïve hSMC, which was significantly greater than the induction by CM of VEH-SMC (2.3-fold; Figure 5a). In all cases, CM from EARLY or END models had no visible effect on SMC gross morphology (Figure 5b).

## 4. Discussion

AAA is a significant cause of morbidity and mortality. With surgical intervention at a critical AAA size being the only treatment option and associated mortality reportedly increasing during the COVID-19 pandemic [39], reliance on surgical repair alone is far from ideal. Exploring novel cell-based interventions that might retard or halt aneurysmal development at an earlier time-point carries considerable promise. A key outcome of the current study was the discovery of a *hyperproliferative* EARLY CCE-SMC phenotype that we propose is an adaptive mechanism attempting to preserve tissue integrity following disturbances in mechanical forces perceived by the SMC. This proposition is upheld by studies that reported the success of SMC seeding in stabilising and slowing AAA progression in a rat xenograft model [40,41,42]. Evidence for an early proliferative phase in AAA growth comes also from our work in an AngII murine model using a PET-CT radiotracer (18F-FLT), in which peak uptake of the tracer (and hence proliferation) was at mid-stage development and not at end-stage [43]. However, prolonged hyperproliferation would ultimately drive premature replicative senescence, thus contributing to vessel weakening and consequent rupture. Indeed, we have consistently observed increased senescence in SMC cultured from both human AAA and porcine END CCE (12-day bioreactor) tissue [23,33]. In this scenario, vascular wall integrity is likely to be irreversibly compromised. The rate of AAA expansion has recently been shown to be slower than previously thought, further widening a therapeutic window where cell-based intervention might be possible [44,45]. Importantly, in order to devise any such therapy, a thorough understanding of SMC behaviour throughout the course of aneurysm development is essential. The current study adds further weight to our previous work on the usefulness of the porcine bioreactor model for studying temporal changes in SMC throughout the evolution of aneurysmal-like lesions.

Aortic rupture is inevitably preceded by mechanical failure. Given our previous findings in SMC cultured from the END porcine model, we tested the mechanical properties of intact END vessels. END CCE vessels exhibited reduced stiffness, strain, and strength when compared to FRESH vessels in both the longitudinal and circumferential directions, concomitant with loss of elastin and SMC content. Whilst EARLY CCE tissue also exhibited a loss of elastin, there was no visible reduction in α-SMA-positive cells. SMC cultured from tissue retrieved after 3 days, whilst morphologically indistinguishable, were hyperproliferative compared to VEH control cells.

The functional roles of SMC in AAA are discrepant. SMC have been reported to confer both a protective, paracrine effect on AAA progression and also exacerbate pathological characteristics by contributing to elements of the ECM synthesis/proteolysis imbalance. For example, evidence of SMC phenotypic switching was observed soon after porcine pancreatic elastase-induced AAA formation in mice [46]. In another study, prevention of SMC phenotypic switching by conditional deletion of Kruppel-like factor 4 appeared to protect against aneurysm formation [47]. Taken together, data from these studies and those described in a recent comprehensive review [48] strengthen the hypothesis that SMC play critical roles in AAA development. Whilst loss of SMC is a major contributor to mechanical failure in aneurysmal tissue [49], this was not clearly apparent in EARLY CCE tissue, leading us to hypothesise that compensatory SMC mechanisms may have masked any obvious changes in gross tissue appearance.

Inflammatory cells are major sources of MMPs [50], although SMC likely contribute to ECM proteolysis via elevated secretion of MMP-2 and -9 [35,51] and hence are implicated in the imbalance in tissue degradation and synthesis mechanisms in AAA. In humans, however, it is clearly not possible to measure MMP expression locally or temporally throughout AAA development. Whilst circulating levels of MMP-2 can readily be measured [52], they do not necessarily reflect local cellular concentrations, and hence correlation with aneurysm size is inappropriate [53]. Given that we observed temporal differences in SMC proliferation between EARLY and END models, our ex vivo model provides a unique opportunity to explore how and when MMP-2 might contribute to SMC dysfunction. Our data showed conclusively that MMP-2 is downregulated at end stage, akin to what was observed in end-stage human AAA-SMC. The EARLY model provides clues that MMP-2 dysregulation in SMC may commence at an early stage of aneurysm development, although a profile of MMP-2 secretion over a wider range of time-points is necessary.

In the current study, END CCE-SMC secreted paracrine factors that were mitogenic to naïve, non-aneurysmal human SMC. This is not without precedent: a senescence-associated secretory phenotype (SASP) is widely documented in the development of atherosclerosis, a pathology that shares mechanisms common to AAA at the level of the SMC [54,55,56]. Whilst this paracrine mechanism is likely to be relevant to all vascular SMC types, irrespective of the species or source, it would be appropriate to affirm by applying conditioned medium to naïve porcine arterial SMC. The mitogenic properties of conditioned media that were observed in this study may well be indicative of accelerated ageing and SASP. Certainly, rhomboid, multinucleated cells in both END CCE and human end-stage AAA-SMC are reminiscent of a prematurely aged phenotype [57], raising the question “what are the cellular and molecular characteristics that define SMC during the intervening phases from early-to-late aneurysm development?” (Figure 6). Studies with additional, sequential time-points (spanning 3–12 days) in bioreactor culture will be necessary in order to delineate pivotal and time-dependent adaptations in tissue strength, as well as in SMC phenotype and function. In view of changes in arterial stiffness in the progression of AAA, an investigation of endothelial-dependent and -independent arterial dilatation using myograph studies on FRESH, EARLY, and LATE tissue may provide additional insights. Taken together, the described studies increase the potential to expose novel targets for further validation, which in turn could reveal therapeutic options for selectively targeting SMC in AAA. Evidence to support this proposition was presented in a recent study, whereby local tyrosine kinase inhibition reduced SMC proliferation and restored a differentiated, contractile phenotype in animal tissues and, importantly, in AAA patient-derived cells [58]. Recent bioinformatics studies have employed transcriptomic and metabolomic analysis of human AAA [59,60,61], although whether these markers might correlate with our porcine data in END-CCE requires investigation. It is feasible that studies of this nature would prove valuable by developing a detailed timeline from which to create new foundations for SMC-specific targets.

The present study provides insights to understanding the progressive nature of AAA development with a particular focus on SMC phenotype and function. Whilst the bioreactor model bridges a gap between in vivo and in vitro models, it nevertheless has limitations. We concur that porcine aorta itself would be the vessel of choice for the study, although size (larger bioreactor required), more variability in vessel dimensions in each chamber, greater thickness (reduced viability, possible necrosis in the inner layers, particularly in END, 12-day cultured vessels), and multiple branch points would require mitigation. Nevertheless, we previously established the advantages of porcine carotid artery in terms of size, thickness, and viability during prolonged culture, and importantly, we justified this approach by demonstrating how closely porcine END-SMC exemplified cells cultured from human end-stage AAA [23,33]. Using contralateral carotid arteries for bioreactor experiments enabled matching of equivalent length and diameter in the two chambers of the bioreactor. The absence of branch points on the carotids facilitated dissection, handling, and assembling, hence minimising physical damage to the tissue during the multiple stages of experimental set-up. In conclusion, in the medium term, this bioreactor holds more potential for exploratory studies ahead of a significant advance to a porcine in vivo model.

## Figures and Tables

**Figure 1 cells-11-01043-f001:**
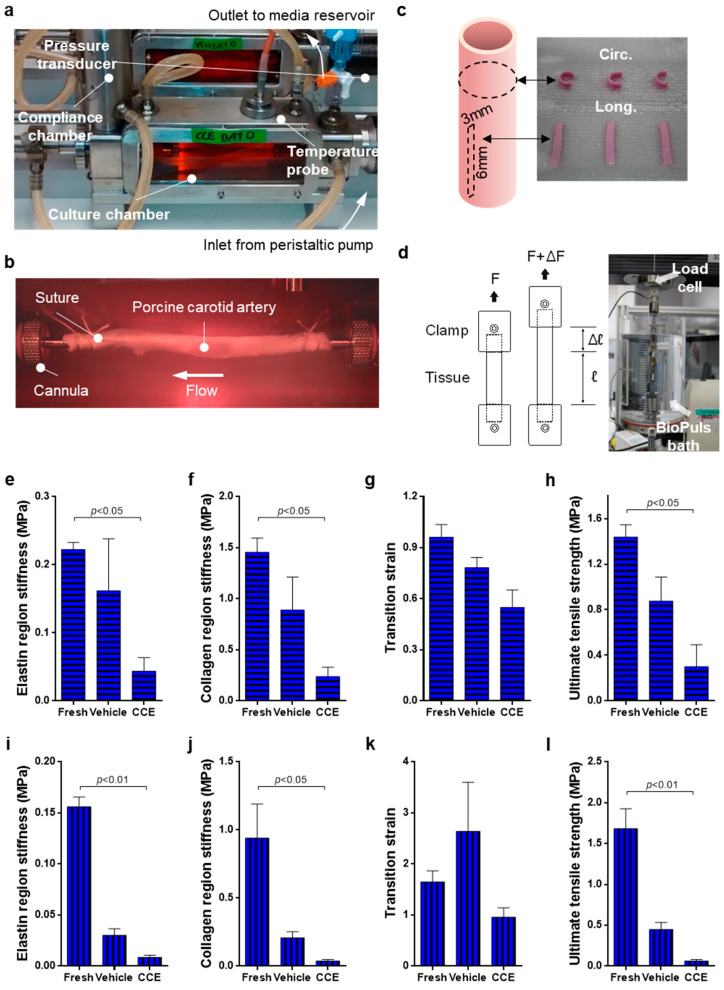
Biomechanical testing of END tissue. (**a**,**b**) VEH and CCE Porcine carotid arteries were mounted within the bioreactor and cultured under constant flow for 12 days; (**c**,**d**) following this, tissue was cut into circumferential and longitudinal strips and underwent biomechanical testing. Experiments were performed on both these tissues and FRESH tissue which had not been cultured in the bioreactor. (**e**) Elastin region stiffness, (**f**) collagen region stiffness, (**g**) transition strain, and (**h**) ultimate tensile strength were recorded for both the circumferential and (**i**–**l**) longitudinal orientations. All *n* = 6 FRESH, *n* = 3 VEH, *n* = 3 CCE, significance assessed using a Kruskal–Wallis test with Dunn’s multiple comparisons test.

**Figure 2 cells-11-01043-f002:**
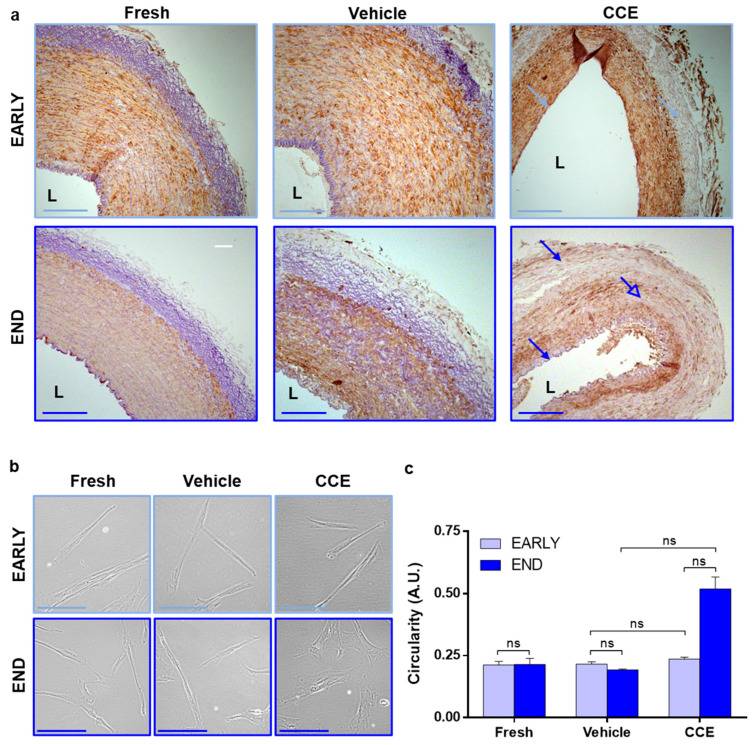
Tissue and cellular morphology from EARLY and END models. (**a**) Tissue was fixed and stained with Miller’s elastin for elastic fibres and α-SMA to identify SMC content. Images captured at ×100 magnification; L = lumen; closed arrows = elastin loss; open arrow = SMC loss; scale bar = 200 µm. (**b**) SMC were isolated from FRESH tissue or after 3 (EARLY) or 12 days (END) in the bioreactor. Images captured at 400× magnification; scale bar = 100 µm. (**c**) The outline of 50 cells per condition per animal was measured, and their circularity calculated. All *n* = 3, significance assessed using the Kruskal–Wallis test and the Mann–Whitney test with Benjamini–Hochberg correction for multiple comparisons. ns: not significant.

**Figure 3 cells-11-01043-f003:**
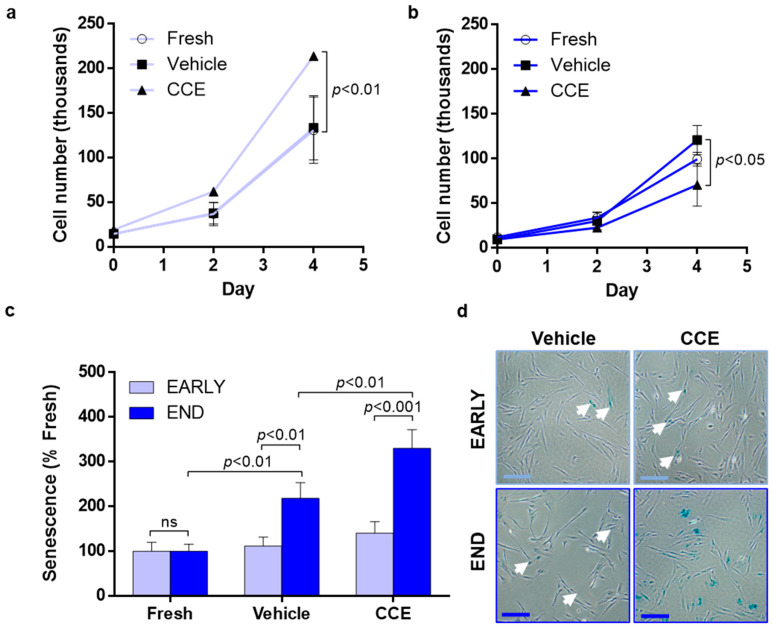
Proliferation and senescence in EARLY and END models. (**a**) EARLY SMC were cultured in full growth medium for up to 4 days, and proliferation was measured using trypan blue exclusion and live cell counts. (**b**) Parallel cultures were maintained and counted from END SMC. (**c**) SMC were cultured for 48 h in full growth medium before being fixed, and senescent cells were detected by positive β-galactosidase staining. (**d**) Representative images taken at ×40 magnification; scale bar = 200 µm, closed arrows indicate areas of positive staining. All *n* = 3, significance assessed using repeated-measures two-way ANOVA with Sidak’s multiple comparison post hoc test.

**Figure 4 cells-11-01043-f004:**
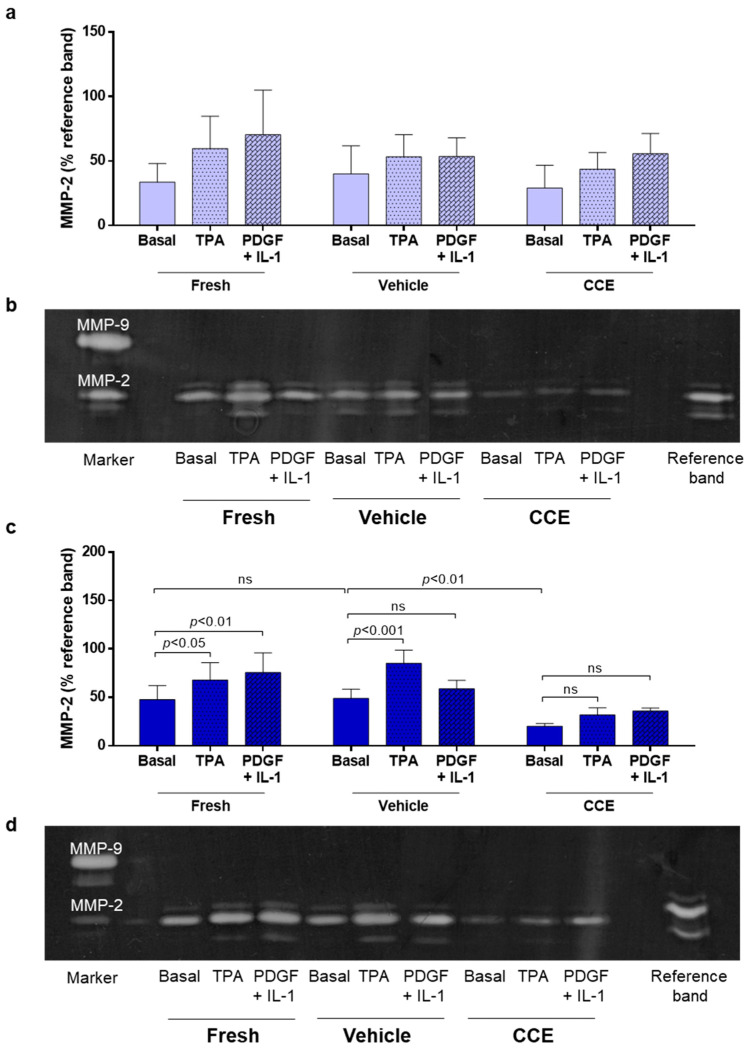
MMP-2 secretion from EARLY and END models. (**a**) EARLY SMC were treated with TPA (100 nM) or PDGF + IL-1α (both 10 ng/mL) and cultured in low serum (0.4%) medium for 48 h. CM was collected and MMP-2 secretion measured using gelatin zymography. (**b**) Representative zymogram. (**c**) Parallel experiments were performed on END SMC. (**d**) Representative zymogram. All *n* = 3, significance assessed using repeated measures two-way ANOVA with Sidak’s multiple comparison post hoc test. ns: not significant.

**Figure 5 cells-11-01043-f005:**
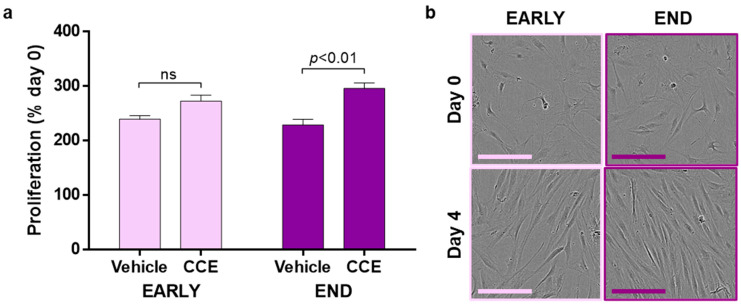
Paracrine modulation of hSMC proliferation. (**a**) CM from EARLY and END SMC were applied to naïve hSMC. Cultures were supplemented with 1% FBS, and proliferation was measured over 4 days. (**b**) Representative images; scale bar = 100 µm; *n* = 4, paired *t*-test. ns: not significant.

**Figure 6 cells-11-01043-f006:**
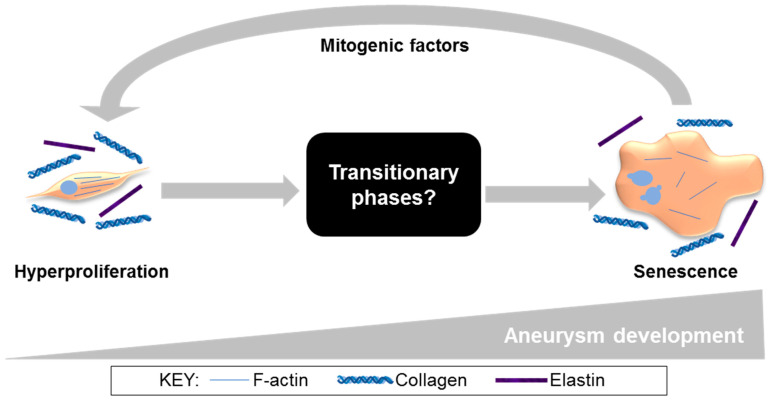
Our proposed model of aneurysm progression, specifically in SMC (from current and published studies) [23,33]. In the EARLY stages of aneurysm development, SMC are spindle-shaped and hyperproliferative. At the END stage, SMC have adopted a rhomboid morphology, disrupted actin cytoskeleton, and characteristics of senescence, secreting paracrine factors that induce proliferation in naïve SMC. Ultimately these mechanisms lead to irreversible SMC dysfunction and loss, vessel wall weakening, and increased risk of aneurysm rupture.

## Data Availability

The data set associated with this article is openly available from the University of Leeds Data Repository.

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
