# Peer review of "Preservation of Smooth Muscle Cell Integrity and Function: A Target for Limiting Abdominal Aortic Aneurysm Expansion?"

_cells, 2022, doi:10.3390/cells11061043_

Round 1

Reviewer 1 Report

This manuscript used ex vivo porcine bioreactor model pre-treated with protease enzyme to create “aneurysm” tissue for understanding SMCs related mechanisms of AAAs.

  1. Statistical analysis stated one-way or two-way repeated measures ANOVA with post-hoc Sidak test or two-sample t-test as appropriate. Suggest that the authors provide statistical method information in each figure. Also suggest that These are parametric tests that should be tested for normal distribution.
  2. The authors state “All n=3 or 4”. These are small sample sizes. Were these biological replicates or technical replicates? The authors should state the information. Also, this is small sample size that may not be appropriate to use parametric tests.
  3. It is unclear whether all studies used carotid arteries and cells from carotid arteries. The authors should clarify. The purpose of this study was to understand mechanisms of AAAs in the aorta. It is unclear about the relevance of using carotid arteries. The limitation of the studies should be discussed.

Author Response

REVIEWER 1

This manuscript used ex vivo porcine bioreactor model pre-treated with protease enzyme to create “aneurysm” tissue for understanding SMCs related mechanisms of AAAs.

  1. Statistical analysis stated one-way or two-way repeated measures ANOVA with post-hoc Sidak test or two-sample t-test as appropriate. Suggest that the authors provide statistical method information in each figure. Also suggest that These are parametric tests that should be tested for normal distribution.

Thank you for these important suggestions. The information regarding statistical tests is now included in individual figure legends, providing greater clarity. We have addressed the point about normal distribution in the response to comment 2 below.

  1. The authors state “All n=3 or 4”. These are small sample sizes. Were these biological replicates or technical replicates? The authors should state the information. Also, this is small sample size that may not be appropriate to use parametric tests.

Thank you, this is a very good point and we appreciate the question of appropriate use of statistical testing throughout our study. Whilst we state “All n=3 or 4” this refers solely to biological replicates (the number of pigs from which vessels were harvested for this study). Within each of those biological replicates are multiple technical replicates. For example, in proliferation studies, cells were isolated from the vessels at the end of each individual bioreactor experiment. All assays were performed in triplicate wells at each time point and from each condition (FRESH, VEH, CCE). This was repeated at passages 3, 4 and 5 to check for consistency of response across passages. Thus, although our “n” number is 3 or 4, this is actually representative of 27 or 36 cell counts in each condition. Similarly for cell circularity, 50 cells from each experiment were measured so our biological n=3 or 4 is representative of either 150 or 200 cell areas. We have used these original datasets to perform normality tests for circularity, proliferation, senescence and MMP secretion.

From these normality tests, all datasets with the exception of “circularity” in CCE-SMC exhibited a normal distribution and thus we can confidently defend retaining parametric tests for those experiments. For circularity, the FRESH and VEH-SMC exhibited a normal distribution but CCE-SMC did not and so these data (originally analysed as a two-way ANOVA) have been reanalysed using a non-parametric Kruskal-Wallis test with Mann-Whitney post-test and a Benjamini-Hochberg correction for multiple comparisons (Benjamini Y, Hochberg Y. Controlling the false discovery rate: A practical and powerful approach to multiple testing. Journal of the Royal Statistical Society Series B (Methodological) 1995;57(1):289-300.

The biomechanics datasets represent a biological n=3 or n=6 and thus are too small to perform normality testing. In this regard we reanalysed all the biomechanics data using non-parametric Kruskal-Wallis test with Dunn’s multiple comparison (original analysis was a one-way ANOVA).

All figures have now been updated with appropriate statistical analyses described above. Discussion of these in the text has been amended as necessary.

  1. It is unclear whether all studies used carotid arteries and cells from carotid arteries. The authors should clarify. The purpose of this study was to understand mechanisms of AAAs in the aorta. It is unclear about the relevance of using carotid arteries. The limitation of the studies should be discussed.

Thank you. In this study all the experiments were performed on porcine carotid arteries and SMC derived from these vessels. The exception was one set of experiments exploring the mitogenic potential of conditioned medium from porcine VEH- and CCE-SMC on human vascular SMC proliferation.

We have published two previous studies using porcine bioreactor-maintained carotid arteries and derived cultured cells, both of which are cited in the current manuscript (J Transl Med 2013; J Vasc Res 2018) [23, 33]. In these earlier manuscripts we have shown the benefits of porcine carotid artery in terms of size, thickness and viability during prolonged culture. Being able to use both carotids for each experiment helped in selecting equivalent length and diameter in each chamber of the bioreactor. Another benefit is that there are no branch points on the carotid, making dissection, handling and assembling more straightforward and avoiding the risk of leakage.  Importantly in [23] we demonstrated how closely porcine END-SMC represented human end-stage AAA.

We concur that porcine aorta would be the vessel of choice for the study, but size (larger bioreactor required), variable vessel size in each chamber, greater thickness (potential compromised viability/necrosis in the inner layers, especially in the LATE vessels) and multiple branch points would need to be considered as confounding issues. We have added new text to the end of the discussion as acknowledgement of the benefits and drawbacks.

Reviewer 2 Report

The authors of the manuscript have investigated an ex vivo model with porcine arteries to explore the biofunction alterations of SMCs in human AAA  development. Overall, the readers may find significance from the model and discoveries, I just have some minor questions:

  1. In the methods, the authors used carotid arteries, but the abdominal artery SMCs' embryo origin is different from carotids, why not use porcine abdominal aortic rings for the study?
  2. What are the biological and experimental replicates of the experiments?
  3. For human SMCs, the authors also used vein cells, which is of the same concern. Whether it could really reflect the molecular and cellular characteristics of AAA SMCs?
  4. Regarding the dynamic blood flow and aorta stifness, it'll be nice to observe EC dependent and EC independent arterial dilatation with Myograph studies on fresh and day 12, which is relevant to AAA development.

Author Response

REVIEWER 2

The authors of the manuscript have investigated an ex vivo model with porcine arteries to explore the biofunction alterations of SMCs in human AAA development. Overall, the readers may find significance from the model and discoveries, I just have some minor questions:

  1. In the methods, the authors used carotid arteries, but the abdominal artery SMCs' embryo origin is different from carotids, why not use porcine abdominal aortic rings for the study?

Thank you. We have published two previous studies using porcine bioreactor-maintained carotid arteries and derived cultured cells, both of which are cited in the current manuscript (J Transl Med 2013; J Vasc Res 2018) [23, 33]. In these earlier manuscripts we have shown the benefits of porcine carotid artery in terms of size, thickness and viability during prolonged culture. Being able to use both carotids for each experiment helped in selecting equivalent length and diameter in each chamber of the bioreactor. Another benefit is that there are no branch points on the carotid, making dissection, handling and assembling more straightforward and avoiding the risk of leakage. Importantly in [23] we demonstrated how closely the porcine carotid tissue and END-SMC represented human end-stage AAA.

We concur that porcine aorta would be the vessel of choice for the study, but size (larger bioreactor required), variable vessel size in each chamber, greater thickness (potential compromised viability/necrosis in the inner layers, especially in the LATE vessels) and multiple branch points would need to be considered as confounding issues. We have added new text to the end of the discussion as acknowledgement of the benefits and drawbacks.

  1. What are the biological and experimental replicates of the experiments?

We have now included this information throughout the Methods section (please see also response to reviewer 1, point 2).

  1. For human SMCs, the authors also used vein cells, which is of the same concern. Whether it could really reflect the molecular and cellular characteristics of AAA SMCs?

In the single set of experiments using human venous SMC, our purpose was not to explore molecular and cellular characteristics of AAA SMC themselves. Instead, we were exploring whether conditioned medium from the porcine cells had mitogenic properties on naïve SMC from any source. On reflection, it would also be useful to repeat these experiments using freshly isolated porcine SMC as the “acceptor” cells for application of conditioned medium. Some additional text to this effect has been added to the Discussion (final paragraph, lines 417-420).

  1. Regarding the dynamic blood flow and aorta stiffness, it'll be nice to observe EC dependent and EC independent arterial dilatation with Myograph studies on fresh and day 12, which is relevant to AAA development.

Thank you for this useful suggestion; we agree that it would be both interesting and informative. This idea could be investigated in a future study and we have added text to the revised “Discussion” section of the manuscript (final paragraph, lines 423-426).

Round 2

Reviewer 1 Report

No further comments